

# Conserved DNA motifs in the type II-A CRISPR leader region

Mason J. Van Orden[*], Peter Klein[*], Kesavan Babu, Fares Z. Najar and Rakhi Rajan

Department of Chemistry and Biochemistry, University of Oklahoma, Norman, OK, USA
[*] These authors contributed equally to this work.

## ABSTRACT

The Clustered Regularly Interspaced Short Palindromic Repeats associated (CRISPR-Cas) systems consist of RNA-protein complexes that provide bacteria and archaea with sequence-specific immunity against bacteriophages, plasmids, and other mobile genetic elements. Bacteria and archaea become immune to phage or plasmid infections by inserting short pieces of the intruder DNA (spacer) site-specifically into the leader-repeat junction in a process called adaptation. Previous studies have shown that parts of the leader region, especially the 3′ end of the leader, are indispensable for adaptation. However, a comprehensive analysis of leader ends remains absent. Here, we have analyzed the leader, repeat, and Cas proteins from 167 type II-A CRISPR loci. Our results indicate two distinct conserved DNA motifs at the 3′ leader end: ATTTGAG (noted previously in the CRISPR1 locus of *Streptococcus thermophilus* DGCC7710) and a newly defined CTRCGAG, associated with the CRISPR3 locus of *S. thermophilus* DGCC7710. A third group with a very short CG DNA conservation at the 3′ leader end is observed mostly in lactobacilli. Analysis of the repeats and Cas proteins revealed clustering of these CRISPR components that mirrors the leader motif clustering, in agreement with the coevolution of CRISPR-Cas components. Based on our analysis of the type II-A CRISPR loci, we implicate leader end sequences that could confer site-specificity for the adaptation-machinery in the different subsets of type II-A CRISPR loci.

## INTRODUCTION

The Clustered Regularly Interspaced Short Palindromic Repeats (CRISPR) and CRISPR-associated (Cas) proteins constitute an RNA-based adaptive immune system that protects bacteria and archaea against phages and mobile genetic elements (*Marraffini & Sontheimer, 2010*; *Sorek, Lawrence & Wiedenheft, 2013*; *Mojica et al., 2005*; *Barrangou et al., 2007*; *Makarova et al., 2006*). CRISPR-Cas systems inactivate intruder DNA, RNA, or both based on the sequence similarity of small CRISPR RNAs (crRNAs) to the invading genetic element and thus protect microbes from phage infections and horizontal gene transfer (*Sorek, Lawrence & Wiedenheft, 2013*; *Marraffini & Sontheimer, 2009*; *Marraffini & Sontheimer, 2010*; *Marraffini & Sontheimer, 2008*; *Brouns et al., 2008*; *Hale et al., 2009*; *Abudayyeh et al., 2016*). The CRISPR-Cas systems are classified into six types (I to VI) with several subtypes within each type based on Cas protein composition (*Abudayyeh et*

Corresponding author
Rakhi Rajan, r-rajan@ou.edu

al., 2016; Makarova & Koonin, 2015; Makarova et al., 2015). An individual bacterium can have multiple CRISPR loci belonging to different CRISPR types. Though types I and III share certain similarities in the overall mechanism of action including crRNA association with multiple Cas proteins (Jackson & Wiedenheft, 2015; Reeks, Naismith & White, 2013; Makarova et al., 2011; Koonin & Makarova, 2013; Shmakov et al., 2015), types II, V, and VI use a single multi-domain protein (Cas9, Cpf1, or C2c2 respectively) along with cognate RNA components for activity (Abudayyeh et al., 2016; Jinek et al., 2012; Jinek et al., 2014; Nishimasu et al., 2014; Zetsche et al., 2015). Cas9 along with a guide RNA is widely being used for genome editing applications, and is being pursued for gene therapy and gene regulation applications (Sternberg & Doudna, 2015; Sontheimer & Barrangou, 2015).

The CRISPR genomic locus is present in either the chromosomal or plasmid DNA as a recurring array of "repeat" and "spacer" units, both of which usually range from approximately 24 to 50 nucleotides (Sorek, Lawrence & Wiedenheft, 2013; Mojica et al., 2005; Ishino et al., 1987; Jansen et al., 2002). Repeats consist of palindromic DNA sequences, while spacers are derived from invader genetic material, and experimental evidence supports the role of spacers in conferring sequence-specific resistance against bacteriophages (Barrangou et al., 2007; Brouns et al., 2008) and plasmids (Marraffini & Sontheimer, 2008). The cas genes that code for the CRISPR system's essential protein components are often located close to the CRISPR locus (Jansen et al., 2002; Haft et al., 2005). The CRISPR leader located 5′ to the first repeat consists of an A/T-rich region around 100–500 nucleotides long with embedded transcriptional promoters (Pougach et al., 2010; Wei et al., 2015; Yosef, Goren & Qimron, 2012). In certain CRISPR types, a 3–5 nucleotide long region present in the invading DNA (Protospacer Adjacent Motif (PAM)) is crucial for Cas proteins to differentiate self from non-self DNA (Sorek, Lawrence & Wiedenheft, 2013; Mojica et al., 2009; Shah et al., 2013), where as in other types this is defined by the crRNA-DNA base pairing patterns (Marraffini & Sontheimer, 2010).

There are three stages in CRISPR mediated defense: adaptation (acquiring new spacers), crRNA biogenesis, and interference (Sorek, Lawrence & Wiedenheft, 2013; Amitai & Sorek, 2016). The adaptation process differs between CRISPR types. The proteins, Cas1 and Cas2, are universally present and essential for adaptation in most of the CRISPR types (Yosef, Goren & Qimron, 2012; Nunez et al., 2014). In certain type II subtypes, Csn2 and Cas4 are also implicated in adaptation along with Cas1 and Cas2 (Barrangou et al., 2007; Garneau et al., 2010). The new spacers are inserted at the leader-repeat junction in most systems, although some variability have been observed in Sulfolobus solfataricus (Erdmann & Garrett, 2012). A minimum of one repeat along with the leader region can promote spacer insertion and in certain CRISPR subtypes, one of the strands of the double-stranded intruding DNA is preferred for spacer acquisition (Wei et al., 2015; Diez-Villasenor et al., 2013). A new spacer insertion is always accompanied by repeat duplication and the first repeat serves as a template for new repeat synthesis (Yosef, Goren & Qimron, 2012). Even though recognition of the PAM sequence by Cas9 is essential for acquisition and insertion of spacers in the correct orientation in vivo in type II CRISPR systems (Wei, Terns & Terns, 2015; Heler et al., 2015; Paez-Espino et al., 2013), it was recently demonstrated that Cas1

and Cas2 from *Streptococcus pyogenes* can specifically integrate spacers into the leader-repeat junction based solely on intrinsic sequence specificity of the first repeat (*Wright & Doudna, 2016*).

*Streptococcus thermophilus* (Sth) DGCC7710 is a model organism widely used for studying various CRISPR processes. Sth DGCC7710 has a total of four CRISPR loci in its genome, and loci CRISPR1 and CRISPR3 belong to type II-A (*Sapranauskas et al., 2011*; *Horvath & Barrangou, 2010*; *Carte et al., 2014*). It was experimentally shown that in Sth DGCC7710, both CRISPR1 and CRISPR3 are active in acquiring new spacers in relation to a new phage or plasmid threat, with CRISPR1 being more active due to the higher frequency of new spacer insertion in this locus compared to CRISPR3 (*Barrangou et al., 2007*; *Wei et al., 2015*; *Garneau et al., 2010*; *Paez-Espino et al., 2013*; *Sapranauskas et al., 2011*; *Carte et al., 2014*; *Deveau et al., 2008*; *Horvath et al., 2008*; *Lopez-Sanchez et al., 2012*). A phylogenetic analysis showed that the repeat and *cas* genes segregate specifically with the locus type (CRISPR1 *vs* CRISPR2 *vs* CRISPR3) in streptococci and in several bacteria belonging to different genera (*Horvath et al., 2008*). Several studies pointed to the indispensability of the 3′ end of the leader in CRISPR adaptation (*Wei et al., 2015*; *Yosef, Goren & Qimron, 2012*; *Erdmann & Garrett, 2012*; *Yosef et al., 2013*; *Jansen et al., 2002*; *Li et al., 2014*; *Lillestol et al., 2009*; *Bernick et al., 2012*; *Erdmann, Le Moine Bauer & Garrett, 2014*). In the CRISPR1 locus of Sth DGCC7710 (type II-A), a *cis*-acting element at the 3′ end of the leader (ATTTGAG) was shown to be essential for adaptation and this region is conserved in several type II-A systems (*Wei et al., 2015*). In *Escherichia coli* BL21 strain, a type I-E CRISPR locus showed defective adaptation following deletions or mutation in the 60 nucleotides towards the 3′ end of the CRISPR leader (*Yosef, Goren & Qimron, 2012*). A pair of inverted repeat regions in the first repeat along with the leader end sequence is critical for adaptation in the type IB system of *Haloarcula hispanica* (*Wang et al., 2016*). Recently, a study in type I CRISPR systems identified leader DNA sequences that are specifically recognized by the integration host factor (IHF) protein to facilitate leader-proximal spacer integration (*Nunez et al., 2016*). It was later shown, however, that under in vitro conditions type II systems integrate spacers specifically into the leader-proximal regions by Cas1–Cas2 activity alone, without the participation of another protein (*Wright & Doudna, 2016*). This highlights the differences in the mechanism of spacer integration between CRISPR types and the possibility of divergent contributions of the leader and repeat sequences in type-specific adaptation.

In order to identify the leader and repeat DNA sequence conservations that may contribute to site-specific spacer integration in type II-A CRISPR systems, we report the analysis of the leader-repeat region belonging to 167 type II-A CRISPR loci from 50 different genera. Eighty-seven of the 167 loci have the 3′ leader end conserved as ATTTGAG (Group 1), 55/167 loci have their 3′ leader end conserved as CTRCGAG (Group 2), and 25/167 possess a CG conservation at the 3′ end of the leader (Group 3). Previous studies that established the importance of ATTTGAG and ACGAG leader end sequences in adaptation of Sth DGCC7710 and *S. pyogenes* (*Wei et al., 2015*; *Wright & Doudna, 2016*), respectively, point to the functional significance of the conserved DNA motifs. A detailed analysis of the Cas proteins associated with the 167 type II-A loci shows protein sequence specificities

that delineate these proteins into groups that mirror the leader end conservation. Thus, our study establishes distinct sub-group specific DNA sequence conservation patterns in the type II-A CRISPR leader that extends across many diverse bacteria demonstrating the ubiquitous nature of the 3′-leader end conservations that were previously observed only in related streptococcal species.

## METHODS

### Processing of genomic data

In this study, the type II-A loci were collected in multiple ways. Initially, Bacterial Generic Feature Format (GFF) and accessioned protein product FASTA files were downloaded from NCBI and scanned for II-A specific Cas protein names (Cas9/Csn1 and Csn2) in the annotation field. The genomes containing Cas9/Csn1 and/or Csn2 annotation entries were downloaded from NCBI in GenBank format. The datasets were screened manually for the presence of *cas1*, *cas2*, *cas9*, and *csn2*, and only the loci with all four type II-A specific *cas* genes were used for further analysis. The genomic region flanking downstream of the *csn2* gene was further processed to extract the leader and the first repeat of the CRISPR array. The protein sequences of Cas9, Cas1, Cas2, and Csn2 that were coded by the upstream region flanking the *csn2* gene were extracted from NCBI. The presence of all four proteins limits our dataset to strictly type II-A loci. A total of 129 loci were identified based on Cas9/Csn1 and/or Csn2 annotation search. Previously, Chylinski et al. reported type II-A loci based on a Cas9 sequence search (*Chylinski et al., 2014*; *Fonfara et al., 2014*). A total of 32 type II-A loci that represented species and genera that were absent in our initial dataset were selected from these reports for our study. In addition, we performed protein sequence homology search by DELTA-BLAST (*Boratyn et al., 2012*) using a representative Csn2 sequence from each subfamily as mentioned in *Chylinski et al. (2014)* (NCBI protein accession number: 116101487 for subfamily I, 116100822 for subfamily II, 389815356 for subfamily III, 385326557 for subfamily IV, 315659845 for subfamily V). By this search a total of 6 loci were identified from bacterial genera *Weissella*, *Globicatella*, *Nosocomiicoccus*, *Caryophanon* and *Virgibacillus*. The final dataset consisted of 167 type II-A loci with a wide representation based on the current knowledge of type II-A diversity. A total of 50 different bacterial genera were present in our dataset (Tables S1 and S2).

The orientation of the Cas proteins was used in assessing the transcription direction of the leader-repeat units. To analyze leader and repeat sequences, an approximately 400-nucleotide stretch of sequences downstream of *csn2* gene were examined using a CRISPR finder tool (*Grissa, Vergnaud & Pourcel, 2007*), and an in-house script to locate the tandem repeats. Since there were differences in the repeat length as it exists in the genomic locus and as reported in the CRISPRdb (*Grissa, Vergnaud & Pourcel, 2007*), we used the in-house program to locate the repeats (Table S3). The accuracy of the repeat extracted by our script was validated manually by checking the genomic data for the length and sequence of the repeat within a CRISPR array. The loci that lacked predicted repeats or Cas protein(s) were omitted from further analysis. In the case of bacteria with multiple CRISPR types, the components belonging to a particular type II-A locus were taken as one dataset. For

example, Sth DGCC7710 has four CRISPR loci. Only loci 1 and 3 that correspond to type II-A were selected for our analysis. The Cas proteins and leader-repeat elements of CRISPR1 were kept as one unit, while those belonging to CRISPR3 represented another unit. Recently, several bioinformatics tools for the identification and analysis of leader and repeat regions have been developed (*Alkhnbashi et al., 2016*; *Biswas et al., 2016*). For a selected subset, we compared the orientation of leader sequences and repeats as predicted by CRISPRDetect tool (*Biswas et al., 2016*) and our results, and saw agreement between the methods.

## Sequence alignment

We used MUSCLE with its default settings (*Edgar, 2004*) to perform all the sequence alignments in this study. The MUSCLE output was used to generate phylogenetic trees with MEGA6 (*Tamura et al., 2013*) using the Maximum Likelihood Tree option and Jones-Taylor-Thornton (JTT) model. Additionally, MUSCLE alignments were used to generate alignment figures in UGENE (*Okonechnikov et al., 2012*) and sequence logos with WebLogo (*Crooks et al., 2004*).

# RESULTS

## Analysis of the 3′ end of the leader

An initial sequence alignment of the last 20 nucleotides of the leader plus the first repeat showed that the 167 loci clustered into distinct groups. These groups had recognizable conservation at the last 7 nucleotides of the 3′ end of the leader and the first 4 nucleotides of the 5′ end of the first repeat, or the leader-repeat junction. To obtain an unbiased separation of the different groups, a Cas1 phylogenetic tree was constructed based on protein sequence similarity. The loci belonging to the different clades of the Cas1 tree were grouped together and a sequence alignment of the last 20 nucleotides of the leader along with the first repeat was performed. In order to facilitate interpretation of the trees and alignments, a smaller representative sample of 60 loci was used to generate the main figures and show the relevant relationships. Figures incorporating all of the 167 loci can be found in the Supplementary Data. Each of the 3 groups was aligned separately to discern the level of conservation within each group (Figs. 1 and 2 and Fig. S1). Strict conservation is seen at the 3′ end of the leader as well as at the 5′ end of the repeat. Group 1 has the 3′ leader end conserved as ATTTGAG (Fig. 1) and Group 2 has the 3′-leader end conserved as CTRCGAG (where R represents a purine) (Fig. 2A). Group 3 has a shorter two nucleotide conservation of CG at the 3′ leader end. In Groups 1 and 2, the last three nucleotides are conserved as GAG (Fig. 2B). An A-rich region is partially conserved adjacent to the CG leader end of Group 3. Interestingly, the CRISPR1 locus of Sth DGCC7710 has the 3′ leader end conserved as ATTTGAG while the CRISPR3 locus has the 3′ leader end conserved as CTACGAG. Of the type II-A CRISPR loci analyzed, 87 belonged to Group1, 55 belonged to Group 2, and 25 belonged to Group 3. Out of the 50 genera analyzed, Group 2 consists of only 5 genera (*Streptococcus, Enterococcus, Listeria, Lactobacillus* and *Weissella*) while Group 1 is much more diverse with 42 different genera. Group 3 accounts for 7 genera, but has many loci belonging to the order Lactobacillales. The leader-repeat junction of Groups 1 and 2 is conserved as GAG/GTTT while in Group 3 it is weakly conserved as CG/GTTT.

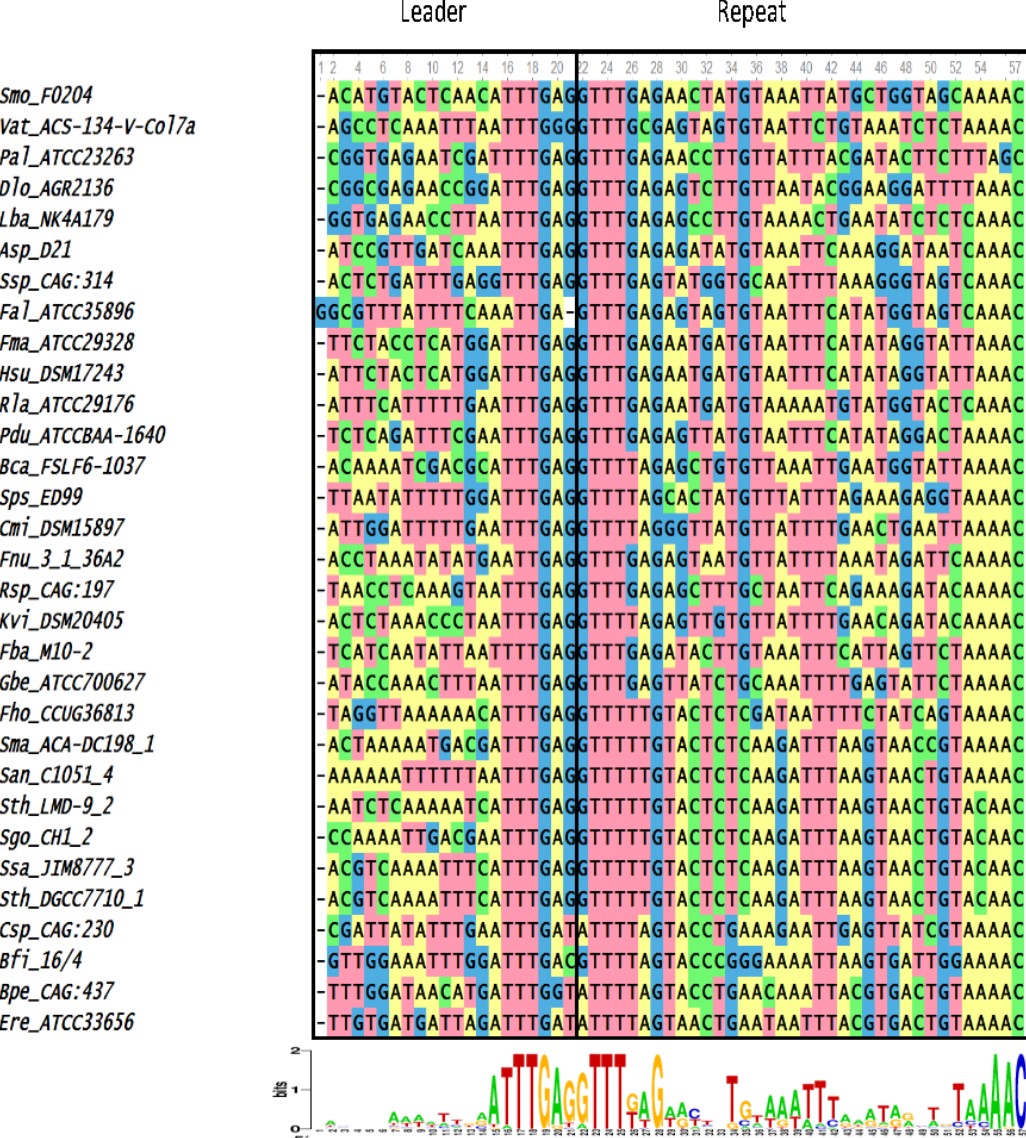

**Figure 1  Sequence alignment of the last 20 nucleotides of the 3′ end of the leader and the first repeat of selected Group 1 species.** Height of the letters in the WebLogo indicates the degree of conservation at specific nucleotide locations. The leader-repeat end is conserved as ATTTGAG/GTTT.

## Analysis of the repeat region

The length of the repeat for the type II-A loci analyzed was 36 nucleotides except in 4 cases (*Enterococcus hirae* ATCC 9790 (35 nucleotides long), *Fusobacterium* sp. 1_1_41FAA (37 nucleotides long), *Lactobacillus coryniformis* subsp. coryniformis KCTC 3167 (37 nucleotides long), and *Lactobacillus sanfranciscensis* TMW 1.1304 (35 nucleotides long)). The first repeat sequences of the 3 groups do not possess any distinguishable motifs that corresponded to the segregation of the different groups (Fig. 3 and Fig. S2). There is a strong sequence conservation at the 5′ end of the repeat as GTTT in all the type II-A loci analyzed (Fig. 3 and Fig. S2). Groups 1 and 2 also share a conserved AAAC motif at the

## A

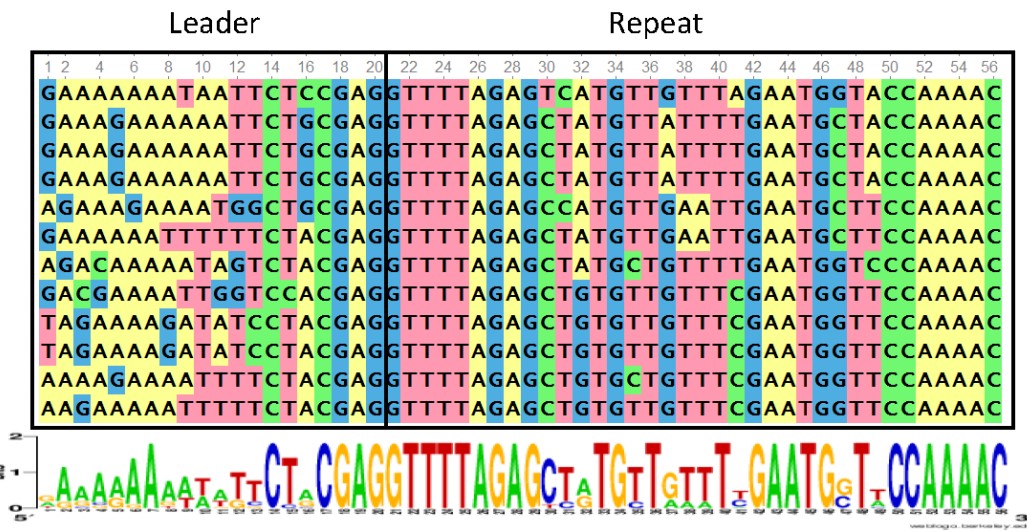

## B

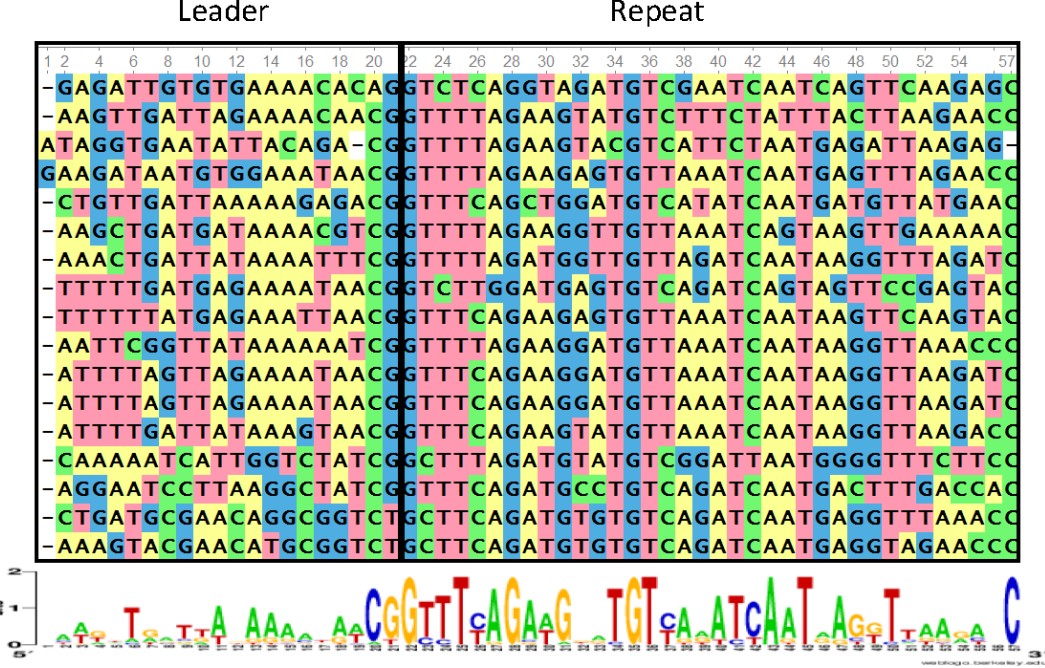

**Figure 2** **Sequence alignment of the last 20 nucleotides at the 3′-end of the leader and the first repeat of selected Group 2(A) and Group 3(B) species.** Height of the letters in the WebLogo indicates the degree of conservation at specific nucleotide locations. The leader-repeat of Group 2 loci is conserved as CTRCGAG/GTTT, where R represents a purine base. For Group 3 members, this region is conserved as CG/GTTT.

3′ end of the repeat. Group 3 members have a conserved C at the 3′ end of the repeat, along with a less conserved A-rich region ahead of the C. The repeat sequence belonging to the Group 2 loci is highly conserved across the entire length of the repeat, which may be attributed to the limited number of genera (5) comprising this group compared to Group 1 (42). In all the type II-A loci analyzed, the first and last nucleotides of the first repeat are conserved as G and C respectively. A phylogenetic tree was generated using the first repeat sequence of the type II-A loci (Fig. 3 and Fig. S2). Even though the reliability of branching is low due to the short length of the sequence, the branches segregate such that members within a clade have similar repeat and leader end conservations. Recently, it was suggested that sequences at the 5′ and 3′ ends of the repeat in *S. pyogenes* type II-A system could be the motifs recognized by Cas1 during spacer acquisition (*Wright & Doudna, 2016*) Hence, the conserved 5′ and 3′ repeat ends observed in the first two groups might indicate type II-A specific repeat ends that are essential for adaptation. Further experimental studies will be required to analyze whether the loosely conserved sequences at the 3′ end of the repeat impact effective adaptation in Group 3. The similarity at the 5′ and 3′ ends of the repeat in the different sub-groups of type II-A system and the fact that exchanging leader ends between CRISPR1 and CRISPR3 loci in Sth DGCC7710 (*Wei et al., 2015*) impaired adaptation shows that the specificity within the sub-groups of type II-A CRISPR system is most probably attributed by the 3′ leader end and not specified by the repeat ends.

## Analysis of Cas proteins

We extended our analysis to verify whether the different groups of type II-A CRISPR loci observed based on the 3′ leader end conservation relates to Cas proteins. The protein sequences of Cas1 belonging to the selected type II-A loci were aligned by MUSCLE and a phylogenetic tree was generated (Fig. 4 and Fig. S3). The loci segregated into 4 main branches, with each branch carrying distinct groups based on the 3′ leader end sequence conservation. A sequence alignment of the leader-repeat junction of the different branches show how the Cas1 sequence is highly correlated with the leader-repeat junction. This confirms previous findings that all the CRISPR-Cas components have coevolved together (*Horvath et al., 2008*). The phylogenetic tree shows that Group 1 loci are very distant in lineage, which later evolved into different subsets with specific leader-repeat-Cas1 combinations. Group 2 and Group 3 evolved for very specific genera, while Group 1 has accommodated divergent genera.

A similar analysis was done for the Cas2, Csn2, and Cas9 proteins. The sequence alignments generated using the sequences of the corresponding Cas proteins were used to build phylogenetic trees (Figs. 5, 6 and Figs. S4, S5). All the clades in the different trees have similar 3′-leader ends, except for a few differences in the Cas9 phylogenetic tree where some Group 3 members appeared along with Group 1. A closer analysis of the sequences showed high variability in the Cas9 lengths, including an extremely short Cas9 sequence (Plo NGRI0510Q) in the outliers, which may have contributed to the random placement of this Cas9 protein. Cas9 also showed a branch (1b) for Group 1 that did not show prominent leader end conservation as what was observed in branch 1a. Except for the few differences in Cas9, our results indicate the presence of distinct groups within the type II-A CRISPR

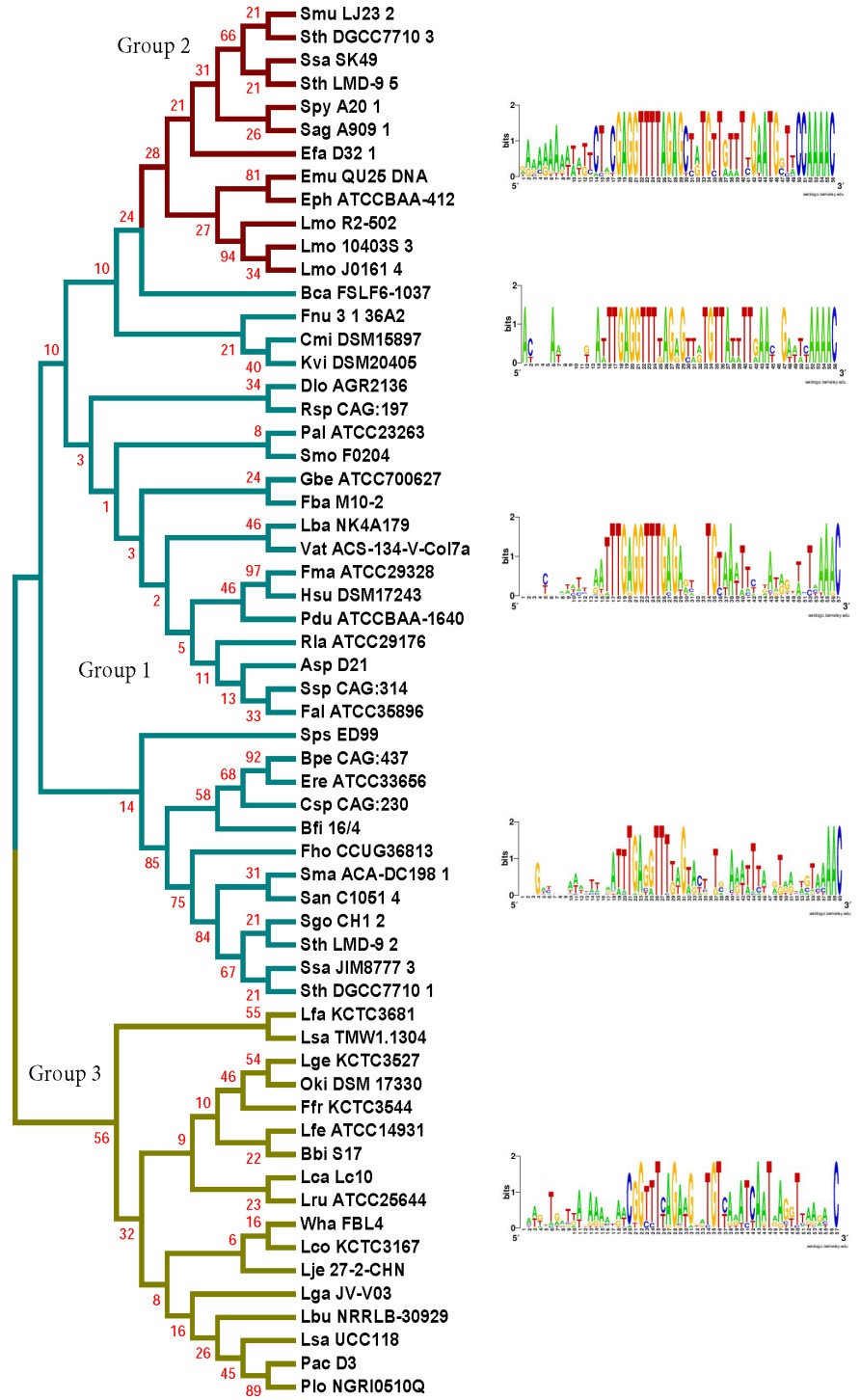

**Figure 3** **Phylogenetic tree generated from the sequence alignment of the first repeats from selected type II-A species.** Groups based on the segregation of the Cas1 tree are shown in cyan (Group 1), red (Group 2), and yellow (Group 3). The tree segregates into 5 main clades and WebLogos were produced with alignments of the last 20 nucleotides at the 3′-end of the leader and the first repeat from the loci within each corresponding branch.

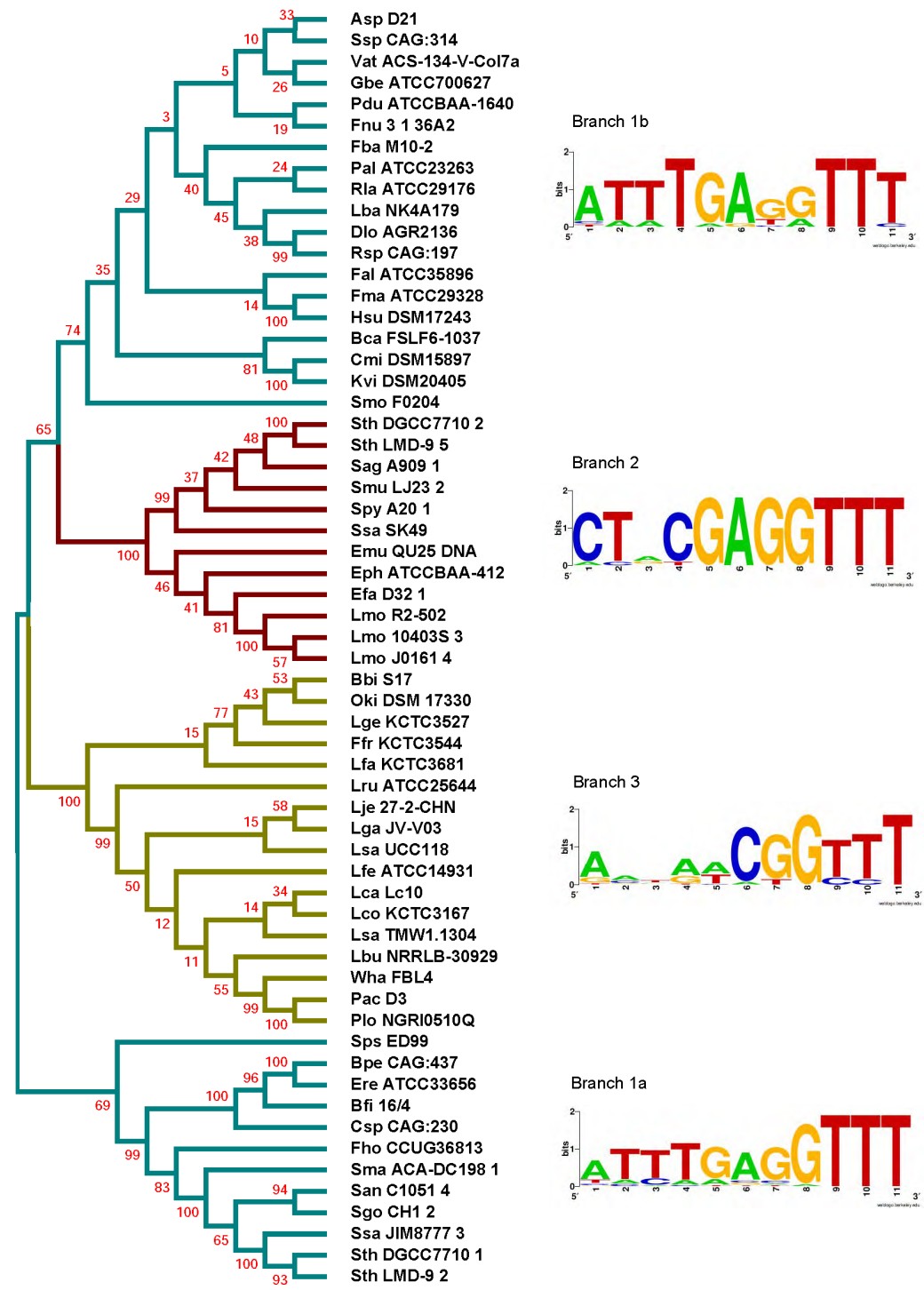

**Figure 4   Phylogenetic tree generated from the sequence alignment of Cas1.** Groups are shown in cyan (Group 1), red (Group 2), and yellow (Group 3). WebLogos were generated by aligning the last 7 nucleotides of the leader and the first 4 nucleotides of the repeat from the loci within each corresponding branch. The tree segregates into 4 branches, two branches showing the Group 1 leader end motif, one branch showing the Group 2 motif, and one branch showing the less-conserved Group 3 leader end. Sps ED99 segregated independently from the final branch but was used in the final branch WebLogo construction based on the leader end and protein length.

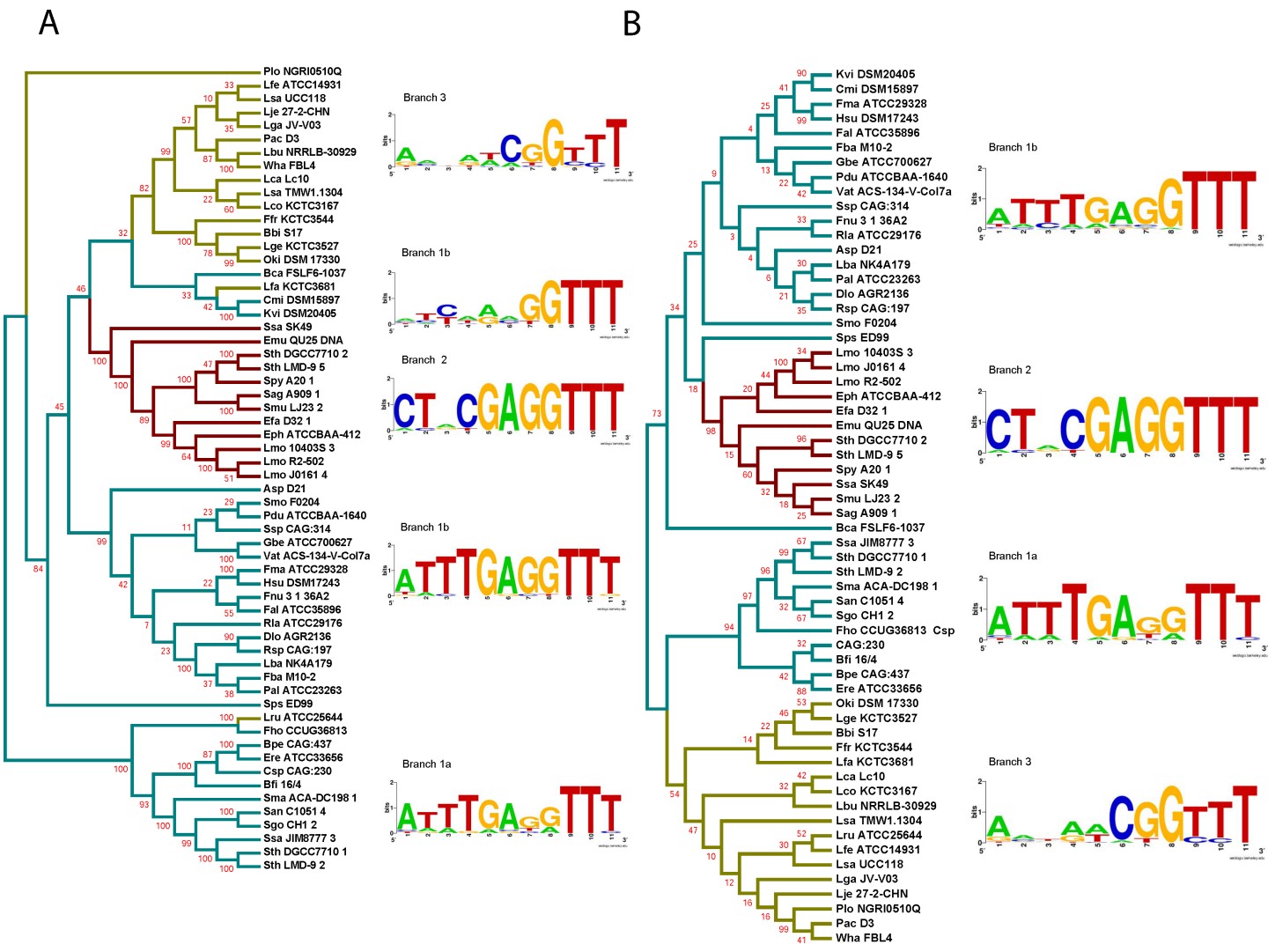

**Figure 5** **Phylogenetic analysis of Cas9 and Cas2.** (A) Phylogenetic tree generated from the sequence alignment of Cas9. Groups based on the segregation of the Cas1 tree are shown in cyan (Group 1), red (Group 2), and yellow (Group 3). The tree shows 5 different branches with two branches showing the Group 1 leader end motif, one branch showing the Group 2 motif, and one branch representing the less-conserved Group 3 leader end. One of the branches represent a very loosely conserved Group 1 loci. Three members of Group 3 segregated away from the normal cluster, of which Plo NGRI0510Q has a very short Cas9 sequence. Lru ATCC25644 and Lfa KCTC3681 have normal length Cas9 sequences. (B) Phylogenetic tree generated from the sequence alignment of Cas2. All the four branches segregate similarly to those of Cas1 phylogenetic tree. WebLogos for both panels of the figure were generated by aligning the last 7 nucleotides of the leader and the first 4 nucleotides of the repeat from the loci within each corresponding branch.

systems that possess conserved 3′ leader ends and group-specific Cas proteins. It was proposed earlier that the longer version of Csn2 evolved first and the shorter Csn2 proteins evolved from the longer versions (*Chylinski et al. 2014*). Interestingly, our phylogenetic analysis agrees with this and shows a branch that represents the ancestor with an average Csn2 length of 320 amino acids (Fig. 6). Three main branches evolved from the ancestor and all of them have an average amino acid length of 218–230 amino acids, but varying 3′ leader ends (Table S4). Thus, the ATTTGAG motif is ancestral and universal in the type

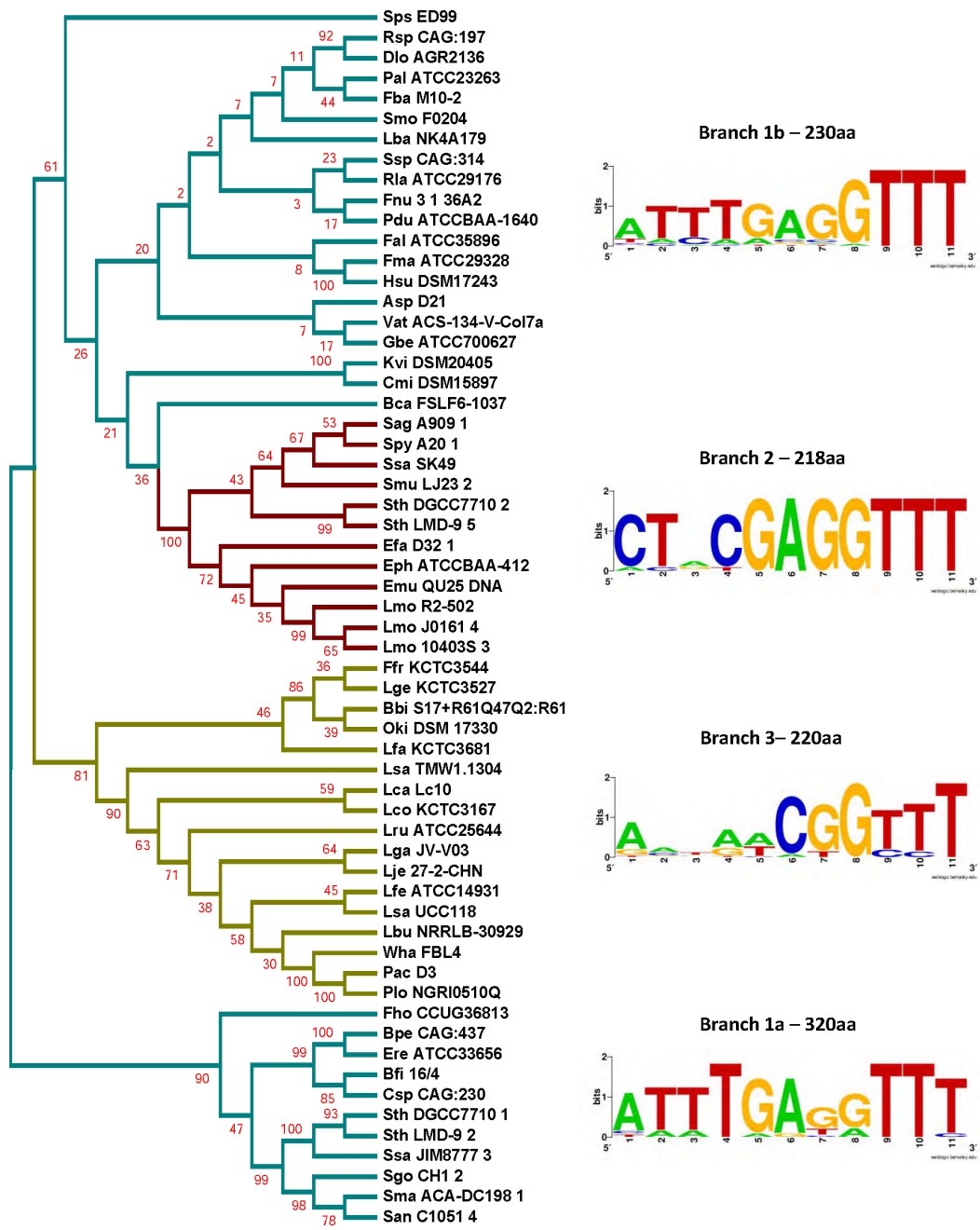

**Figure 6** **Phylogenetic tree generated from the sequence alignment of Csn2.** Groups based on the segregation of the Cas1 tree are shown in cyan (Group 1), red (Group 2), and yellow (Group 3). WebLogos were generated from aligning the last 7 nucleotides of the leader and the first 4 nucleotides of the repeat from the loci within each corresponding branch. Values next to branch labels indicate the average length of the proteins (in amino acids, aa) within the branch. Two branches show the Group 1 leader end motif, one branch shows the Group 2 motif, and one branch shows the less conserved Group 3 leader end.

II-A systems, which later developed to have a similar (ATTTGAG), deviating (CTRCGAG), or less conserved (CG) 3′ leader end, with a corresponding change in the protein sequences of all four type II-A Cas proteins. Examining the lengths of Cas1, Cas2, and Cas9 from different groups, we did not observe a strong correlation between the average length of these Cas proteins and the branching group that they belonged.

## DISCUSSION

Though previous studies have shown that the leader-repeat region is important for adaptation, the specific features of the leader-repeat region that may recruit Cas1–Cas2 for adaptation are not clearly defined. We focused on the sequence conservation around the leader-repeat junction and found three distinct DNA motifs at the 3′ leader ends; Group 1 (ATTTGAG), Group 2 (CTRCGAG), and Group 3 (CG). The presence of a conserved 3′ leader end, despite a low sequence conservation in the upstream regions of the leader in bacteria belonging to 50 different unrelated genera, strongly suggests that these DNA motifs play a role in site-specific adaptation. One of the most interesting observations from this analysis is the conservation of GAG/GTTT as the leader-repeat junction in both Group 1 and Group 2 (82%, 117 out of 142 loci) of the type II-A system.

Several studies have implicated the importance of the leader and repeat sequences to drive faithful adaptation. Terns and coworkers (*Wei et al., 2015*) reported that streptococci with repeats similar to that present in the CRISPR1 locus (Group 1) of Sth DGCC7710 have the 30 leader end conserved as ATTTGAG. The accompanying experimental work clearly demonstrated that the 10 nucleotides present at the 3′ end of the leader and the first repeat are essential and sufficient for adaptation, even in a non-CRISPR locus (*Wei et al., 2015*). It was concluded that sequences at the leader-repeat junction recruits the adaptation machinery to this region for integration of new spacers (*Wei et al., 2015*). In a recent study that analyzed the spacer variation in 126 human isolates of *S. agalactiae*, the 3′ leader end of most of the isolates had a TACGAG sequence (*Lier et al., 2015*). Our analysis that focused on many divergent genera uncovered that the DNA motifs that were previously known to be important for streptococcal adaptation is in fact more ubiquitous and conserved across different bacteria.

The importance of the sequences of the leader and the first repeat in driving adaptation is conserved across different CRISPR types. The 60 nucleotides towards the 3′ end of the type I-E CRISPR locus of *E. coli* is essential for adaptation (*Yosef, Goren & Qimron, 2012*). The disruption of the first repeat sequence that left the stem-loop structure intact prevented successful adaptation in a type IE system, leading to the conclusion that the cruciform structure of the repeat alone is not sufficient for adaptation (*Arslan et al., 2014*). Another study showed that the −2 (second last position of leader) and +1 (first nucleotide of repeat) positions of leader-repeat regions are crucial for adaptation in *E. coli* (type IE) and *S. solfataricus* (type I-A) (*Rollie et al., 2015*; *Nunez et al., 2015*) Other studies have experimentally demonstrated that leader and repeat sequences are important for adaptation in streptococcal type II-A systems corresponding to Groups 1 and 2 that we identified in our study (*Wei et al., 2015*; *Wright & Doudna, 2016*). Comparing our results

with the earlier studies show that leader-repeat sequence conservation that we observed in type II-A sub-groups is relevant for adaptation across diverse bacteria.

There is an interplay between the leader and repeat sequences in adaptation that is CRISPR type-specific. For example, in the type I-B system of *H. hispanica*, inverted repeats (IR) present within the first repeat are essential for recruiting the adaptation machinery to the leader-repeat junction. Once the IRs are located within a repeat, a cut is made by the Cas1–Cas2 complex at the leader-repeat junction and the sequence of the leader is critical for this step. The second cut at the repeat-spacer end is based on a ruler-mechanism and does not depend on the sequence of the repeat (*Wang et al., 2016*). Whereas, in a type II-A system corresponding to our Group 2, it was shown that both the repeat-spacer and the repeat-leader ends could be cleaved by Cas1-Cas2 and that for a faithful adaptation the leader-repeat junction is essential (*Wright & Doudna, 2016*). In the Group 1 type II-A locus of Sth DGCC7710, mutations in the last 10 nucleotides of the leader abolished adaptation (*Wei et al., 2015*). This study also elegantly showed that substitution of the 10 nucleotides at the 3′ end of Group 1 leader with that of Group 2 leader abolished adaptation following a phage challenge, further emphasizing the importance of the locus specific leader-repeat junction in adaptation (*Wei et al., 2015*). Thus, our observation of the group-specific sequence conservation in type II-A systems at the leader end, along with a lack of distinct group-specific motifs at the 5′ and 3′ ends of the first repeat, shows that the sub-group specificity in type II-A adaptation arises from the leader sequences that might be specifically distinguished by the Cas1–Cas2 proteins belonging to each sub-group.

Both Groups 1 and 2 are active for adaptation and interference (*Barrangou et al., 2007*; *Garneau et al., 2010*; *Paez-Espino et al., 2013*; *Sapranauskas et al., 2011*; *Carte et al., 2014*; *Deveau et al., 2008*; *Horvath et al., 2008*; *Lopez-Sanchez et al., 2012*), while Group 3 has been shown to be active in DNA interference (*Sanozky-Dawes et al., 2015*). Introduction of the type II-A Group 2 locus into *E. coli* protected the bacterium from phage and plasmid infection (*Sapranauskas et al., 2011*), demonstrating that intrinsic specificities of protein and DNA components of a CRISPR sub-type are sufficient to drive adaptation and there are no organismal requirements. The three different DNA motifs that we observed at the 3′-end of the leader of the type II-A CRISPR loci may represent three specific functional adaptation units, perhaps guided by leader-sequence specific Cas protein(s). The third group, which consists mostly of lactobacilli, with only two nucleotides conserved instead of seven nucleotides at the 3′ leader end in Groups 1 and 2 may represent a more diverse adaptation complex where the protein-DNA sequence interactions are not as tight. It was noted recently that there is considerable variation in the spacer content, even in the ancestral spacers, in *L. gasseri* strains that indicates considerable divergence between the strains (*Sanozky-Dawes et al., 2015*), thus accounting for the low level of sequence conservation at the 3′-end of the leader. This study also showed that the spacers matched plasmids and temperate phages, though it is not clear how *L. gasseri* acquires spacers from prophages that do not pose a threat to bacterial survival (*Sanozky-Dawes et al., 2015*). These environmental factors may contribute to the low sequence conservation at the 3′-end of the leader in Group 3. Further experiments will be required to assess the adaptation process in Group 3. Group 3 could also be a result of an insufficient amount of genomic data

available to completely resolve any more conserved motifs hidden in the different leader end sequences found within the group.

Repeat sequences are specific to a CRISPR locus, even within sub-types (*Horvath et al., 2008*; *Chylinski et al., 2014*). The first two nucleotides of the first repeat was shown to be essential for adaptation in the CRISPR1 locus of Sth (*Wei et al., 2015*) and the first six nucleotides are essential in adaptation in *S. pyogenes* (*Wright & Doudna, 2016*). The importance of G as the first nucleotide in the repeat for efficient disintegration reaction was demonstrated for both *E. coli* and *S. solfataricus* Cas1 proteins (*Rollie et al., 2015*). We found conservation at the ends of the repeat between groups (Fig. 3). Only 17/167 loci analyzed did not possess a GTTT at the 5′-end of the first repeat, and 3/167 of the loci did not possess a conserved C at the 3′ end. It was previously reported that purified *E. coli* Cas1 possesses nuclease activity against several types of DNA substrates including single stranded DNA, replication forks, Holliday junctions etc. without adequate intrinsic sequence specificity and that the four-way DNA junctions recruits Cas1 protein (*Babu et al., 2011*). Recently, more studies point to the importance of DNA sequence specificity, especially at the 3′-end of the leader, for driving Cas1 for adaptation (*Wei et al., 2015*; *Wang et al., 2016*; *Rollie et al., 2015*). The essentiality of IHF for site-specific adaptation in type I-E indicates that even though Cas1 may have the ability of non-sequence specific cleavage in certain CRISPR types, tight regulation by other cellular proteins may enhance site-specific spacer insertion. The position of the IHF site is 9–35 nucleotides upstream of the leader-repeat junction in type I systems (*Nunez et al., 2016*). The 20 nucleotides of the 3′ leader end that we analyzed for the type II-A did not possess any similarity to the IHF binding site. It is possible that a cruciform structure formed by leader-repeat or repeat palindromic regions along with specific leader-repeat sequences may recruit the Cas1–Cas2 complex for spacer insertion and that this requirement is critical under in vivo conditions.

All four Cas proteins are essential for successful adaptation in vivo in type II-A systems (*Barrangou et al., 2007*; *Yosef, Goren & Qimron, 2012*; *Heler et al., 2015*). Previous studies have shown that the CRISPR components and Cas proteins have coevolved (*Horvath et al., 2008*). Our analysis showed that all the four type II-A specific Cas proteins and the first repeat clustered into identical groups with similar 3′ leader ends. Even though Cas1 protein sequences within type II-A are highly conserved, there are certain differences that segregate them into distinct groups and interestingly these groups have distinct leader sequence conservation. It was previously reported that type II-A CRISPR systems have distinct operon organization that correlates with Csn2 sequence, making Csn2 the signature protein for type II-A systems (*Chylinski et al., 2014*). The longer version of Csn2 originated first and the shorter version evolved from the longer version (*Chylinski et al., 2014*). Our analysis shows that the length of Csn2 is conserved across different clusters (Fig. 5). Looking at Fig. 6, branch 1a segregated early from the rest of the tree and consists of the longer version of Csn2, while branches 1b, 2, and 3 all consist of the shorter version of Csn2. Correlating Csn2 branching to the leader end sequences, it is evident that our Group 1 motif of ATTTGAG is present in the ancestral strains, which later evolved to distinct sub-groups possessing either Group 1, Group 2 (CTRCGAG) or Group 3 (CG) leader ends.

## CONCLUSION

We present an extensive bioinformatic analysis of type II-A CRISPR systems spanning 50 different bacterial genera. We demonstrated the ubiquitous nature of two distinct DNA motifs at the 3′ end of the leader: Group 1 (ATTTGAG) and Group 2 (CTRCGAG) and also discovered a new group (Group 3) with a limited sequence conservation at the 3′-end of the leader. The leader-repeat junction is highly conserved for Groups 1 and 2 as GAG/GTTT. Our work proposes that the Cas proteins of each sub-group within the type II-A system should make sequence-specific association with its cognate DNA region for successful spacer insertion. The observations further strengthen the previous notion that a highly specific interplay between Cas proteins and cognate leader-repeat regions is essential for effective adaptation (*Wei et al., 2015*; *Yosef, Goren & Qimron, 2012*; *Diez-Villasenor et al., 2013*; *Arslan et al., 2014*; *Rollie et al., 2015*).

## ACKNOWLEDGEMENTS

We thank Sungho Suh for help with genomic data collection and processing.

### Funding

The research reported in this paper was supported by an Institutional Development Award (IDeA) from the National Institute of General Medical Sciences of the National Institutes of Health under the grant number P20GM103640. The funders had no role in study design, data collection and analysis, decision to publish, or preparation of the manuscript.

### Grant Disclosures

The following grant information was disclosed by the authors:
Institutional Development Award (IDeA): P20GM103640.

### Competing Interests

The authors declare there are no competing interests.

### Author Contributions

- Mason J. Van Orden and Peter Klein conceived and designed the experiments, performed the experiments, analyzed the data, wrote the paper, prepared figures and/or tables, reviewed drafts of the paper.
- Kesavan Babu and Fares Z. Najar performed the experiments, analyzed the data, reviewed drafts of the paper.
- Rakhi Rajan conceived and designed the experiments, performed the experiments, analyzed the data, contributed reagents/materials/analysis tools, wrote the paper, reviewed drafts of the paper.

### Data Availability

The raw data has been supplied as a Supplementary File.

## Supplemental Information

Supplemental information for this article can be found online at http://dx.doi.org/10.7717/peerj.3161#supplemental-information.

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
