# Peer review of "Conserved DNA motifs in the type II-A CRISPR leader region"

_PeerJ, doi:10.7717/peerj.3161_

## Round 0.1 · original submission · Minor Revisions

Please address the minor and major points raised by the two reviewers. Particularly, you should pay attention to the comment of Reviewer 1 concerning the alignments shown in Figure 1: " but group one contains many sequences (the first 7 or 8) that do not conform to the ATTTGAG consensus. It would be better to leave those that do not conform out of each of the main alignments."

Reviewer 1 ·

Basic reporting

The paper describes and analysis of the leader-first repeat region of CRISPR-Cas type-II-A CRISPR arrays. The work is entirely bioinformatic analysing a set of 167 sequences selected to represent diversity in the type II-A systems. The work appears to be competently done and makes some interesting findings, mainly confirming and extending prior observations.

Experimental design

The motifs are described as belonging to type II-A. The authors should provide better evidence that the dataset does not contain other subtypes in addition to type II-A, either type-II or other types. This is particularly important for the newly proposed group 3 sequences.
The approach used to make a multiple alignment fixed on the leader/repeat boundary and then assess what the end of the leader was is quite straightforward.

Validity of the findings

The finds appear valid for type II-A, provided these leader/repeats are only being processed by type II-A systems (see above).

The alignments shown in Figure 1 are appropriate for most members of groups 2 and 3, but group one contains many sequences (the first 7 or 8) that do not conform to the ATTTGAG consensus. It would be better to leave those that do not conform out of each of the main alignments.

Additional comments

Results: The last sentence (line 163) should be in the analysis of the repeat region section and the boundary defined e.g. GAG/GTTT rather than 5’-GAGGTTT-3’. Note the 5’, 3’ on these sequences is not required throughout (unless they are written unconventionally e.g. 3’ to 5’)

Discussion/Introduction
Some highly relevant recent studies have not been included or commented on.
The authors should cite and consider (or exclude) any possible link of these leader sequences to those recently described (Nunez et al. CRISPR Immunological Memory Requires a Host Factor for Specificity. Mol Cell. 2016.).

Other authors have considered sequences at the 5’ end of the repeat discussed here (e.g. GTTTT) and have used them to define and orient repeats, this approach should be commented on and compared here. (Biswas A, et al. Accurate computational prediction of the transcribed strand of CRISPR non-coding RNAs. Bioinformatics. 2014;30(13):1805-13; Biswas A, et al CRISPRDetect: A flexible algorithm to define CRISPR arrays. BMC Genomics. 2016;17(1):356)

Although very recently published, others have also analysed leaders (Alkhnbashi OS, Shah SA, Garrett RA, Saunders SJ, Costa F, Backofen R. Characterizing leader sequences of CRISPR loci. Bioinformatics. 2016;32(17):i576-i85.) and the 5’ ends of repeats. The findings presented here should be compared to those.

All the sequences should be provided as multiFASTA file e.g. for each group 1-3 and the rest in the supplement. This will facilitate further analysis.

Reviewer 2 ·

Basic reporting

The article is generally well-written, with professional English and appropriate references cited throughout. There are a few typos and suggestions to improve clarity of figures (listed below):
1. In line 8, author Klein has a “1” superscript without accompanying footnote.
2. In line 70, “CRISRPR” instead of CRISPR is used
3. In line 98, “sup-group” instead of sub-group is used
4. In line 172, a period is missing in the sentence that ends with “…Fig. S2)”
5. In line 175, “A-region” instead of A-rich region is used
6. In line 176, “Group 2 loci” instead of Group 2 locus is used
7. In line 281, two periods appear at the end of the sentence
8. In Figures 3-6, sequence logos are used to show areas of conservation at leader-repeat junctions. It would be helpful for the reader if it were clearly marked where the leader ends and repeat begins (as was nicely done for the sequence alignments in Figs. 1 and 2).

Experimental design

The authors present a bioinformatics analysis of 167 Type II-A CRISPR loci and identify regions of sequence conservation in the leader and repeats that are likely important for the process of CRISPR-Cas adaptation. It has already been established through experimental means that sequences in the leader and repeats for various systems (including Type II) are important for adaptation. While the work is provides a more comprehensive view through bioinformatics analysis, the authors fail to explicitly define the knowledge gap. This could be corrected by including a transition statement in the abstract and introduction that explains the knowledge gap and rationale for the current work .

One point that requires clarification in the methods section is how organisms with multiple CRISPR-Cas types (and therefore multiple copies of Cas1 and Cas2) handled? For example, S. thermophilus has four distinct CRISPR-Cas systems and four copies each of Cas1 and Cas2. This is not uncommon in distinct organisms. In the different analyses used, were all Cas1 and 2 proteins in an organism considered, or just the one(s) most proximal to the Type II-A CRISPR locus? This information needs to be included in the methods section.

Validity of the findings

There are some areas where data interpretation is unclear and/or not statistically sound:
1. Under the results section titled “Analysis of Cas proteins”, it is unclear whether Cas1 lengths are significantly different from one another given the range of sizes (high and low) shown in Table S4. A statistical analysis needs to be conducted in order to establish a correlation between Cas1 length and repeat groups.
2. In Supplementary Table 4, the Cas2 lengths appear off given the high and low ranges shown. Please check these for accuracy.
3. Under the results section titled “Analysis of the repeat region”, the authors indicate conserved sequences at 5’-end and 3’-ends of repeats as 5’-GTTTT, and 5’-AAAAC; however not all sequence logos in Fig. 3 show all four T’s or A’s conserved in these motifs. Consider changing the consensus sequences to 5’-GTTT and 5’-AAAC, respectively.
4. Again regarding the multiple Cas1 and Cas2 proteins inside cells with distinct CRISPR-Cas systems, possible cross-talk between these proteins and different repeat-leader groups could exist. The impact of this possibility in the context of the current bioinformatics analysis should be discussed.

---

## Round 0.2 · accepted · Accept

The authors properly addressed all the issues raised by the two reviewers